# Polynomial Affine Model of Gravity in Three-Dimensions

Oscar Castillo-Felisola [1,2,*,†], Bastian Grez [1,†], Oscar Orellana [3,†], Jose Perdiguero [1,†], Francisca Ramirez [1,†], Aureliano Skirzewski [4,†] and Alfonso R. Zerwekh [1,2,†]

1 Departamento de Física, Universidad Técnica Federico Santa María, Casilla 110-V, Chile; bastian.grez@sansano.usm.cl (B.G.); jose.perdiguerog@gmail.com (J.P.); temporiones@gmail.com (F.R.); alfonso.zerwekh@usm.cl (A.R.Z.)
2 Centro Científico Tecnológico de Valparaíso, Casilla 110-V, Chile
3 Departamento de Matemáticas, Universidad Técnica Federico Santa María, Casilla 110-V, Chile; oscar.orellana@usm.cl
4 Instituto de Física, Facultad de Ciencias Iguá 4225, esq. Mataojo, Montevideo 11400, Uruguay; askirz@gmail.com
* Correspondence: o.castillo.felisola@pm.me
† These authors contributed equally to this work.

**Abstract:** In this work, we explore a three-dimensional formulation of the polynomial affine model of gravity, which is a model that extends general relativity by relaxing the equivalence principle through the exclusion of the metric from the set of fundamental fields. In particular, in an attempt to gain insight of the role of the torsion and nonmetricity in the gravitational models, we consider homogeneous and isotropic cosmological models, for which their solutions are classified in a decisions tree. We also show a few of these explicit solutions that allow the definition of (alternative/emergent) metrics derived from the connection.

**Keywords:** alternative models of gravity; affine gravity; cosmological models; three-dimensional gravity; exact solutions

**PACS:** 04.20.Jb; 04.50.+h; 04.50.Kd; 98.80.-k

## 1. Introduction

During the first quarter of the twentieth century, Einstein built the general theory of relativity, attempting to render gravitational interactions and the special theory of relativity compatible. The resulting model, known as general relativity, turned to provide a geometrical interpretation of gravitational interactions, where gravitational forces are a manifestation of the curvature of spacetime.

General relativity is so far the most successful model of gravitational interactions at our disposal, explaining a large amount of experimental observations [1]. Nonetheless, the completion of general relativity with the standard model of particles is insufficient to account for effects such as the rotation of the Halos of Galaxies [2–4], and the value of the cosmological constant [5,6]. We should add to that the difficulty the ability to consistently quantize the model [7–14].

In order to bypass these vicissitudes, it is necessary either to modify the matter content of the standard model of particles or to find an alternative model to general relativity. In this work, we focus on the second type of modifications, particularly to enhance the gravitational sector by allowing a general connection and restricting the role of the metric.[1]

Interestingly, a general notion of (continuous and connected) manifolds is determined by the affine connection, which is a geometrical object more fundamental than the metric. A manifold ($M$) equipped with an affine connection ($\hat{\Gamma}$) is called an affine manifold, and it admits the notion of parallel transports. Hence, the Riemannian geometry behind general relativity is a particular case of the above when the connection is symmetric in their lower indices and compatible with a metric tensor field, i.e., $\hat{\Gamma}_{[\mu}{}^{\lambda}{}_{\nu]} = 0$ and $\nabla^{(\hat{\Gamma})}g = 0$. In affinely

connected manifolds, it is possible to define $\binom{0}{2}$ tensors derived from the connection, e.g., the Ricci tensor field, for which its symmetric part might serve as a metric field if it is non-degenerated [15]. Sometimes, these tensors are referred to as connection descendent or induced or emergent metric tensor fields.

From a modern perspective, the idea of formulating gravity as an affine theory is attractive since other fundamental interactions are gauge theories, for which their fundamental fields are gauge connections, and such theories are re-normalizable. Therefore, an affine formulation of gravity would bring the theory to the same footing with gauge theories.

The first proposals of affine gravity were considered by Einstein, Eddington and Schrödinger [16–19], but those models did not offer sufficient phenomenological novelty, while their treatments were significantly more complex. More recent affine models of gravity were proposed by Kijowski and collaborators [20–23], Popławski [24–27], Krasnov and collaborators [28–33], and ourselves [34,35].

Albeit the fact that affine formulations of gravity conceptually differ from general relativity, the predictions obtained from earlier models do not provide significant differences from those of general relativity, while their manipulation was harder.[2] It has been understood that even if two models are equivalent at the dynamical level, their generalisations might be inequivalent [36]. However, subsequent models predict novel effects.

In the four-dimensional version of the polynomial affine model of gravity, even if we restrict ourselves to the torsion-free sector of the model, the vacuum solutions of Polynomial Affine Gravity include the vacuum solutions of general relativity as a subset [35]. Moreover, some vacuum solutions of our affine model account for effects that in general relativity are induced by the presence of matter. This is interpreted as a mimicking of matter effects by the non-Riemannian structure of geometry [37]. Although the underlying geometry of the polynomial affine model of gravity does not require the existence of a metric ab initio, it is possible to define connection-descendent metrics,[3] allowing to distinguish between null and non-null geodesics (self-parallel), which would describe the free-fall trajectories for mass-less and massive particles [38]. What is worth noticing is the existence of a connection-descendent metric permits making contact with other quantities of interest in cosmological and astronomical/astrophysical applications, such as the red shift.

Moved by the moral from the early attempts to build affine models of gravity, it is convenient to analyze the model in lower dimensions, where the number of parameter is in general diminished. These models are a playground to test methods that would be applied in four dimensions, but they also might be applied in other branches of physics since some gauge theories in three dimensions can be interpreted as the high-temperature limit of four-dimensional models [39].

Historically, the three-dimensional version of general relativity was firstly studied by Staruszkiewicz [40], who considered Schwarzschild-like solutions and noticed a relation between the presence of massive point particles and conical singularities. It was soon understood that in three-dimensional general relativity there were no propagating degrees of freedom as a consequence of a relation between the Riemann and Ricci tensors (they are not independent since the conformal Weyl tensor in three dimensions is trivial). Due to this, the interest for three-dimensional gravity decreased. It was only after a series of papers by Deser, Jackiw, and Templeton [41–44], who added a nontrivial topological term to the action, that three-dimensional gravity re-emerged. In those works, the authors showed that the additional term induced a massive model of gravity dubbed Deser–Jackiw–Templeton gravity (or DJT for short).

Later, it was shown that the three-dimensional gravity modified by the Pontryagin–Chern–Simons Lagrangian is equivalent to a Yang–Mills theory (called Chern–Simons gravity), and its perturbative expansion was renormalizable [45].[4] Later, Bañados, Teitelboim and Zanelli found the first black hole solution [47], disproving the triviality of the classical three-dimensional gravity. After these there was a boost in the search of exact solutions [48] and analysis of their quantum aspects [49].

From all of the above, it is possible to conclude that the non-triviality of three-dimensional gravity comes from topological aspects. However, when one considers affinely connected manifolds, it is possible to define the projective Weyl tensor field:

$$W_{\mu\nu}{}^{\lambda}{}_{\rho} = \mathcal{R}_{\mu\nu}{}^{\lambda}{}_{\rho} - \frac{1}{D-1}\left(\mathcal{R}_{\nu\rho}\,\delta^{\lambda}_{\mu} - \mathcal{R}_{\mu\rho}\,\delta^{\lambda}_{\nu}\right),$$

without the use of a metric tensor. The projective Weyl tensor, unlike the conformal Weyl tensor (which requires a metric and has terms, with the scalar curvature), does not vanish necessarily. Therefore, gravitational models involving the affine connection might express dynamical effects unaccounted for by general relativity. An example of the modifications to the dynamics comes from the existence of nonmetricity which, unlike the torsion, introduces a term that is responsible of the non-coincidence of geodesics and self-parallel curves. Moreover, such a quantity cannot be set to zero through a coordinate transformation due to its tensor character, i.e., it is only zero in a frame if it vanishes in any other frame.

The aim of this study is to review some of the aspects of the polynomial affine model of gravity introduced in Ref. [34], focusing our attention on the three-dimensional scenario. In this context, we find ansätze compatible with the cosmological principle, which allows us to solve the field equations. Interestingly, some of the solutions admit non-degenerated derived metrics.

## 2. What Is the Polynomial Affine Model of Gravity?

Our purpose is to build an alternative model of gravity where the degrees of freedom mediating the interaction come from a fundamental affine connection, instead of coming from the metric tensor field. In order to achieve our goals, we suppressed the use of the metric in the formulation of the action and focused on terms that are polynomials in fields (and their derivatives). As a consequence, the number of possible terms involved in the action is finite, as we shall show by an analysis of the index structure later in this section.

Starting from the affine connection, $\hat{\Gamma}_{\mu}{}^{\lambda}{}_{\rho}$, we can decompose it into its irreducible components by separating the symmetric and skew-symmetric parts (in the lower indices),

$$\hat{\Gamma}_{\mu}{}^{\lambda}{}_{\nu} = \hat{\Gamma}_{(\mu}{}^{\lambda}{}_{\nu)} + \hat{\Gamma}_{[\mu}{}^{\lambda}{}_{\nu]} = \Gamma_{\mu}{}^{\lambda}{}_{\nu} + \mathcal{B}_{\mu}{}^{\lambda}{}_{\nu} + \mathcal{A}_{[\mu}\delta^{\lambda}_{\nu]}.$$

In the last equality, we split the skew-symmetric component of the connection further into its trace, $\mathcal{A}$, an its traceless part, $\mathcal{B}$. Hereafter, we denote the symmetric component of the connection by $\Gamma$ (without the hat). It is worth mentioning that the skew-symmetric part of the connection is related to the torsion tensor field; therefore, both $\mathcal{A}$ and $\mathcal{B}$ are the building blocks of torsion.

Our guideline to build the action functional would be the invariance under diffeomorphisms, and the field content of the model would be the symmetric connection ($\Gamma$), the trace of the torsion ($\mathcal{A}$), and the traceless torsion ($\mathcal{B}$). Note that the torsion-descendent fields are tensor fields, while $\Gamma$ is (still) a connection. Consequently, the symmetric connection cannot (in principle) enter directly to the action, but instead it should enter through the covariant derivative, $\nabla$.

In order to build an action functional successfully, we require a volume form in the affine manifold. Such volume form is a generalisation of the well-known Riemannian volume form, which is defined by the metric tensor field, $dV_g = d^3x\,\sqrt{g}$, where symbol $g$ represents the (absolute value of the) determinant of the metric tensor field. We denote the affine volume by $dV^{\mu\nu\lambda} = dx^{\mu} \wedge dx^{\nu} \wedge dx^{\lambda}$.

How do we build the action with these ingredients? Since the action is the integral of the Lagrangian, which is a scalar density, we should write down all possible terms that are polynomial in the fields, which can be contracted to form scalar densities. Since $\mathcal{A}$, $\mathcal{B}$, and $\nabla$ have tensor character and the volume form is the only element with *weight*, we can classify the terms in the action by an analysis of the indices structure.[5]

Note that all "tensor" fields have a simple net lower index (e.g., the $\mathcal{B}$ field has one upper and two lower indices; thus, it would contribute to a net single index), while the volume form has three upper indices. Let us define the *index balance operator* as $\mathcal{N}$, which returns the value $+1$ for each upper index and $-1$ for lower indices, and the *weight operator* as $\mathcal{W}$, which returns the weight of the tensor density. Then, an operator of the following form is obtained:

$$\mathcal{O} = \mathcal{A}^m \mathcal{B}^n \nabla^p \, \mathrm{d}V^q,$$

where the superindices denote the power of the element (number of times it might appear), and it can be part of the Lagrangian when the net index balance is zero and its weight is one; thus, it should satisfy the following:

$$\mathcal{N}(\mathcal{O}) = 3q - m - n - p = 0, \qquad \mathcal{W}(\mathcal{O}) = q = 1,$$

if the operator could take part in the action.

Now that we know the restrictions in the power of the fields, we write down all possible contractions, and use the symmetries of the fields to eliminate redundant terms. The most general action (up to boundary terms) is given by the following (see Ref. [34]):

$$
\begin{aligned}
S = \int \mathrm{d}V^{\alpha\beta\gamma} \Big( & B_1 \, \mathcal{A}_\alpha \mathcal{A}_\mu \mathcal{B}_\beta{}^\mu{}_\gamma + B_2 \, \mathcal{A}_\alpha \mathcal{F}_{\beta\gamma} + B_3 \, \mathcal{A}_\alpha \nabla_\mu \mathcal{B}_\beta{}^\mu{}_\gamma + B_4 \, \mathcal{B}_\alpha{}^\mu{}_\nu \mathcal{B}_\beta{}^\nu{}_\lambda \mathcal{B}_\gamma{}^\lambda{}_\mu \\
& + B_5 \mathcal{R}_{\alpha\beta}{}^\mu{}_\mu \mathcal{A}_\gamma + B_6 \, \mathcal{R}_{\mu\alpha}{}^\mu{}_\nu \mathcal{B}_\beta{}^\nu{}_\gamma + B_7 \, \Gamma_\alpha{}^\mu{}_\nu \partial_\beta \Gamma_\gamma{}^\nu{}_\nu \\
& + B_8 \left( \Gamma_\alpha{}^\mu{}_\nu \partial_\beta \Gamma_\gamma{}^\nu{}_\mu + \frac{2}{3} \Gamma_\alpha{}^\mu{}_\nu \Gamma_\beta{}^\nu{}_\lambda \Gamma_\gamma{}^\lambda{}_\mu \right) \Big),
\end{aligned}
\tag{1}
$$

where the topological Chern–Simons terms have been included explicitly. In the above, we use the symbols $\mathcal{F}$ to denote the *field strength* of the $\mathcal{A}$-field, i.e., $\mathcal{F}_{\beta\gamma} = \partial_\beta \mathcal{A}_\gamma - \partial_\gamma \mathcal{A}_\beta$, and we use $\mathcal{R} = \mathcal{R}^\Gamma$ to denote the curvature of the symmetric connection.

In comparison with the action of general relativity, Equation (1) seems unpleasant at first. However, we should have in mind that the Hilbert–Einstein action is far from being the most general one compatible with the symmetries. For this reason, it is useful to highlight some nice features of the polynomial affine model of gravity, which can be read from its action: (i) Similarly for the other fundamental interactions, the fundamental field is a connection; (ii) all coupling constants are dimensionless, which is desirable from the point of view of Quantum Field Theory, since the superficial degree of divergence vanishes; (iii) at least at a classical level, the model seems to exhibit scale invariance; (iv) the number of possible terms in the action is finite (we usually refer to this property as the rigidity of the model), giving the impression that in the hypothetical scenario of quantisation, the model would be renormalisable, since all counter-terms should have the form of terms already present in the original action.

The field equations derived from the action in Equation (1) are as follows:

$$2B_1 \mathcal{A}_\alpha \mathcal{B}_\nu{}^\alpha{}_\rho + 2B_2 \mathcal{F}_{\nu\rho} + B_3 \nabla_\mu \mathcal{B}_\nu{}^\mu{}_\rho + B_5 \mathcal{R}_{\nu\rho}{}^\mu{}_\mu = 0, \tag{2}$$

$$2B_1 \mathcal{A}_\nu \mathcal{A}_\rho - 2B_3 \nabla_{(\nu} \mathcal{A}_{\rho)} + 3B_4 \mathcal{B}_\nu{}^\mu{}_\sigma \mathcal{B}_\rho{}^\sigma{}_\mu + 2B_6 \mathcal{R}_{\mu(\nu}{}^\mu{}_{\rho)} = 0, \tag{3}$$

$$
\begin{aligned}
B_3 \mathcal{A}_\mu \mathcal{B}_\rho{}^\nu{}_\sigma + B_5 \left( \delta_\mu^\nu \mathcal{F}_{\rho\sigma} + \delta_{[\rho}^\nu \mathcal{F}_{\sigma]\mu} \right) + B_6 \left( 2\delta_\mu^\nu \nabla_\tau \mathcal{B}_\rho{}^\tau{}_\sigma + \delta_\rho^\nu \nabla_\tau \mathcal{B}_\sigma{}^\tau{}_\mu + \delta_\sigma^\nu \nabla_\tau \mathcal{B}_\mu{}^\tau{}_\rho \right) & \\
+ B_7 \left( \delta_\mu^\nu \mathcal{R}_{\rho\sigma}{}^\lambda{}_\lambda + \delta_{[\rho}^\nu \mathcal{R}_{\sigma]\mu}{}^\lambda{}_\lambda \right) + B_8 \left( \mathcal{R}_{\rho\sigma}{}^\nu{}_\mu + \delta_{[\rho}^\nu \mathcal{R}_{\sigma]\lambda}{}^\lambda{}_\mu \right) &= 0. \quad (4)
\end{aligned}
$$

We have to mention that the field equations presented above are obtained by a naive optimization of the action functional, without taking into account the possible existence of second class constraints that could result in the necessity of adding boundary contributions to the action.

In order to solve the field equations, we use the same strategy as in general relativity to provide an ansatz that must be compatible with the requirements of our problem.

### 3. Building the Ansätze

Riemannian geometry is the arena where general relativity lays its grounds, and the fundamental geometrical object in Riemannian geometry is the metric tensor field. Since the metric tensor field is related to the notion of distance, in the easiest examples it is straightforward to propose ansätze. However, for less trivial examples one utilises the notion of the Lie derivative and its relation with the concept of symmetries.

Although the action of the Lie derivative on tensors (and even tensor densities) is commonly known [50], it is less known that it is possible to define the Lie derivative of an affine connection [51–53], given by the following relation,

$$
\begin{aligned}
\pounds_V \hat{\Gamma}_\mu{}^\lambda{}_\nu &= V^\sigma \partial_\sigma \hat{\Gamma}_\mu{}^\lambda{}_\nu - \hat{\Gamma}_\mu{}^\sigma{}_\nu \partial_\sigma V^\lambda + \hat{\Gamma}_\sigma{}^\lambda{}_\nu \partial_\mu V^\sigma + \hat{\Gamma}_\mu{}^\lambda{}_\sigma \partial_\nu V^\sigma + \frac{\partial^2 V^\lambda}{\partial x^\mu \partial x^\nu} \\
&= \hat{\nabla}_\mu \hat{\nabla}_\nu V^\lambda + \hat{\mathcal{R}}_{\rho\mu}{}^\lambda{}_\nu V^\rho - 2\hat{\nabla}_\mu \left( \Gamma_{[\nu}{}^\lambda{}_{\rho]} V^\rho \right).
\end{aligned}
\tag{5}
$$

In the above, the vector field $V$ represents a generator of the symmetry group, i.e., in mathematical terminology, $V$ defines the symmetry flow. Note that the second line is a covariant expression, since the skew-symmetric part of the connection (i.e., the torsion) is a tensor field.

Therefore, the strategy to find ansätze would be the following: (i) Define the desired symmetries in concordance with the physical problem we would like to solve; (ii) obtain the vector fields associated to the generators of the symmetry group and express them in the coordinate system of preference; (iii) restrict the coefficients of the affine connection by solving the set of equations defined by $\pounds_V \hat{\Gamma}_\nu{}^\lambda{}_\rho = 0$. Note that this represents a set of 27 partial differential equations for each vector $V$.

In order to illustrate the procedure, we use spherical coordinates, $(t, r, \varphi)$, and show that the general affine connection is compatible with isotropy and the cosmological principle (i.e., isotropic and homogeneous).

In three-dimensions, the vector fields generating the rotation and translations along a two-dimensional subspace of constant curvature are represented by the following:

$$
\begin{aligned}
J &= \begin{pmatrix} 0 & 0 & 1 \end{pmatrix}, \\
X &= \sqrt{1 - \kappa r^2} \begin{pmatrix} 0 & \cos\varphi & -\frac{1}{r}\sin\varphi \end{pmatrix}, \\
Y &= \sqrt{1 - \kappa r^2} \begin{pmatrix} 0 & \sin\varphi & \frac{1}{r}\cos\varphi \end{pmatrix},
\end{aligned}
\tag{6}
$$

where $\kappa = -1, 0, +1$ is the normalised Gaußian curvature of the two-dimensional subspace.

### 3.1. Isotropic Connection

Straightforwardly, one can observe from Equation (5) that, since the vector field $J$ is constant, the non-homogeneous contribution to the Lie derivative vanishes; thus, it coincides with the Lie derivative of a $\binom{1}{2}$-tensor.

Moreover, the explicit relations derived from the condition of the vanishing Lie derivative of the connection along the vector $J$ are as follows:

$$
\pounds_J \hat{\Gamma}_\mu{}^\lambda{}_\nu = \partial_\varphi \hat{\Gamma}_\mu{}^\lambda{}_\nu = 0.
\tag{7}
$$

The solution is that none of the components of the affine connection depend on the angular coordinate $\varphi$. Unfortunately, this symmetry argument does not kill any of the components of the connection! Consequently, using only an argument of isotropy, it would be almost impossible to find an analogous case of the Schwarzschild black hole—which in three-dimensions is called Bañados–Teitelboim–Zanelli (BTZ) black hole [47,48,54,55].

### 3.2. Isotropic and Homogeneous Connection

In order to find the affine connection compatible with the cosmological principle, we have to solve the conditions of the vanishing Lie derivative of the connection along the vector fields $X$ and $Y$. This procedure is long and tedious but straightforward, and we refer the interested reader to check Refs. [37,56] to see more details.

After solving the conditions, the irreducible components of the affine connection are given by the following:

$$\Gamma_t{}^t{}_t = j(t), \qquad \Gamma_i{}^t{}_j = g(t)S_{ij},$$
$$\Gamma_i{}^k{}_j = \gamma_i{}^k{}_j, \quad \Gamma_t{}^i{}_j = \Gamma_j{}^i{}_t = h(t)\delta_j^i + f(t)S^{ik}\epsilon_{kj}\frac{r}{\sqrt{1-\kappa r^2}}, \tag{8}$$

where $f$, $g$, $h$, and $j$ are functions of time, while $S_{ij}$ and $\gamma_i{}^j{}_k$ are the two-dimensional rank two symmetric tensor and connection compatible with isotropy and homogeneity, defined by the following:

$$S_{ij} = \begin{pmatrix} \frac{1}{1-\kappa r^2} & 0 \\ 0 & r^2 \end{pmatrix},$$

and

$$\gamma_r{}^r{}_r = \frac{\kappa r}{1-\kappa r^2}, \qquad \gamma_\varphi{}^r{}_\varphi = -r(1-\kappa r^2), \qquad \gamma_r{}^\varphi{}_\varphi = \frac{1}{r}, \qquad \gamma_\varphi{}^\varphi{}_r = \frac{1}{r}.$$

If one compares with the Levi-Civita connection obtained from a Friedman–Robertson–Walker metric, one notices that the function $j$ is zero, the function $f$ is something entirely new (and it exists solely in three-dimensions), and the functions $g$ and $h$ are defined in terms of the scale factor $a$, as $g = a\dot{a}$ and $h = \dot{a}/a$.

It is possible to show that in the affine case, a reparametrization of the time allows us to set the function $j = 0$ [57].

The nonvanishing components of the $\mathcal{B}$-field are as follows:

$$\mathcal{B}_\varphi{}^t{}_r = -\mathcal{B}_r{}^t{}_\varphi = \xi(t)\frac{r}{\sqrt{1-\kappa r^2}},$$
$$\mathcal{B}_t{}^r{}_\varphi = -\mathcal{B}_\varphi{}^r{}_t = \psi(t)r\sqrt{1-\kappa r^2}, \tag{9}$$
$$\mathcal{B}_r{}^\varphi{}_t = -\mathcal{B}_t{}^\varphi{}_r = \frac{\psi(t)}{r\sqrt{1-\kappa r^2}},$$

while the nonvanishing component of the $\mathcal{A}$-field is $\mathcal{A}_t = \eta(t)$.

### 4. Cosmological Solutions

Now that we have the general form of the fields, we substitute such ansatz into the field equations in Equations (2)–(4). The resulting system of equations is as follows:

$$2B_8 gf - B_3 \xi \eta = 0, \tag{10}$$
$$B_6(2g\psi + \dot{\xi}) - B_8 fg = 0, \tag{11}$$
$$B_8(2gh + \kappa - \dot{g}) = 0, \tag{12}$$
$$B_3 \eta \psi + B_8(2hf + \dot{f}) = 0, \tag{13}$$
$$B_1 \eta^2 - B_3 \dot{\eta} + 3B_4 \psi^2 - 2B_6(\dot{h} + h^2 - f^2) = 0, \tag{14}$$
$$B_3 g\eta - 3B_4 \psi \xi + B_6(\kappa + \dot{g}) = 0, \tag{15}$$
$$2B_1 \eta \xi + B_3(2g\psi + \dot{\xi}) = 0. \tag{16}$$

Notably, the Equation (10) is algebraic. However, we can obtain other algebraic relations from Equations (11) and (16), and from Equations (12) and (15). These expressions are as follows:

$$2B_8 gf - B_3 \xi \eta = 0, \tag{17}$$

$$\frac{B_8}{B_6} gf + 2\frac{B_1}{B_3} \xi \eta = 0, \tag{18}$$

$$3\frac{B_4}{B_6} \psi \zeta - 2gh - \frac{B_3}{B_6} g\eta = 2\kappa. \tag{19}$$

Equations (17) and (18) can be seen as a system of equations for variables $f$ and $\eta$ as functions of $g$ and $\xi$, but their independence is dictated by the determinant of the following system coefficients,

$$\Omega = B_8 \left[ 4\frac{B_1}{B_3} + \frac{B_3}{B_6} \right] g\xi. \tag{20}$$

A thorough search for solutions requires taking into account all possible cases, which can be represented by decision trees (see Figure 1). We shall sketch the types of solutions in two cases: (i) when the determinant in Equation (20) is nonvanishing, which requires $f = \eta = 0$; and (ii) when the determinant $\Omega$ vanishes, for which the decision tree has many branches.

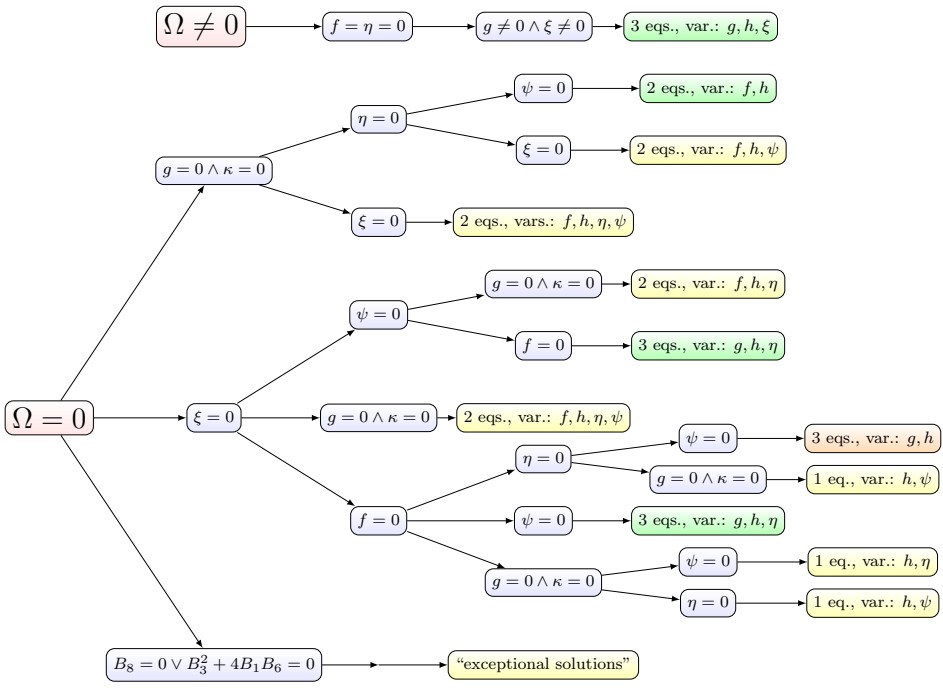

**Figure 1.** Decision tree depicting the scan of cosmological solutions.

Interestingly, from the solutions obtained in the scan, only a few of them have all the fields switched on [57]. Below, we present some of these exceptional solutions.

Firstly, the case with $g = \kappa = \xi = 0$ has as nontrivial field, and the equations of the following:

$$\dot{f} + 2fh = -\frac{B_3}{B_8} \eta \psi, \tag{21}$$

$$\dot{h} + h^2 - f^2 = \frac{1}{2B_6} \left( B_1 \eta^2 - B_3 \dot{\eta} + 3B_4 \psi^2 \right), \tag{22}$$

which are solved when all the functions are inversely proportional to $t$. Note that with this configuration, $\Gamma$, $\mathcal{A}$, and $\mathcal{B}$ are nonvanishing.

Another branch of solutions is found when the coupling constants are not all independent, e.g., for $B_3^2 + 4B_1B_6 = 0$, condition $\Omega = 0$ is satisfied without fixing $g$ or $\xi$. Under these conditions, we observed a couple of exceptional solutions.

First, for $B_1 = B_3 = 0$, the solution is characterized by the following functions:

$$f(t) = 0, \qquad\qquad g(t) = -\kappa t + C_g, \quad h(t) = \frac{1}{t - \kappa C_g},$$

$$\xi(t) = \sqrt{-\frac{2B_6}{3B_4}}C_g, \quad \psi(t) = 0, \qquad\qquad \eta(t) = \text{arbitrary}, \tag{23}$$

for $\kappa \neq 0$. While for $\kappa = 0$, the functions defining the connection are as follows:

$$f(t) = 0, \qquad\qquad g(t) = C_m t + C_g, \qquad\qquad h(t) = \frac{C_m}{2(C_m t + C_g)},$$

$$\xi(t) = \sqrt{-\frac{2B_6}{3B_4}(C_m t + C_g)}, \quad \psi(t) = \sqrt{-\frac{B_6}{6B_4}\frac{C_m}{C_m t + C_g}}, \quad \eta(t) = \text{arbitrary}. \tag{24}$$

Secondly, for $\kappa = 0$, we proposed the ansatz $g(t) = t^n$ with $n \in \mathbb{R} - \{-2, \left[\frac{1-\sqrt{33}}{4}, \frac{1+\sqrt{33}}{4}\right]\}$, with the following condition.

$$B_4 = -\frac{8B_6^3(n+2)}{3B_8^2(2n^3 - 3n^2 - 3n + 4)}.$$

The functions defining the connection are as follows,

$$f(t) = \frac{\sqrt{2n^2 - n - 4}\,\text{sgn}(B_8)}{2t}, \quad g(t) = t^n,$$

$$h(t) = \frac{n}{2t}, \qquad\qquad \xi(t) = \frac{\sqrt{2n^2 - n - 4}\,\text{sgn}(B_8)t^n}{2B_6}, \tag{25}$$

$$\eta(t) = \frac{2B_6}{B_3 t}, \qquad\qquad \psi(t) = \frac{(n-1)\sqrt{2n^2 - n - 4}\,\text{sgn}(B_8)}{4tB_6}.$$

The solutions shown above are interesting because they admit some form of connection-descendent metric. In particular, the solution from Equation (25) admits several emergent metrics, for example, (i) the Ricci tensor field and (ii) the Popławski metric [27],[6] defined from the torsion tensor field as $g_{\mu\nu} = \mathcal{T}_{\mu}{}^{\lambda}{}_{\rho}\mathcal{T}_{\nu}{}^{\rho}{}_{\lambda}$, which are in general non-degenerated. Without further details, for this solution, the metrics are of the following form:

$$\text{Ric} = -\frac{A(n)}{t^2}\,\text{d}t \otimes \text{d}t + nt^{n-1}\left(\frac{\text{d}r \otimes \text{d}r}{1 - \kappa r^2} + r^2\,\text{d}\varphi \otimes \text{d}\varphi\right),$$

and the following as well:

$$\text{Popl} = -\frac{F(n, B_i)}{t^2}\,\text{d}t \otimes \text{d}t + G(n, B_i)t^{n-1}\left(\frac{\text{d}r \otimes \text{d}r}{1 - \kappa r^2} + r^2\,\text{d}\varphi \otimes \text{d}\varphi\right),$$

where $A$, $F$, and $G$ are numerical factors depending on coefficients $B_i$ and the value of $n$ [57].

## 5. Discussion and Concluding Remarks

In this article, we have briefly reviewed the polynomial affine model of gravity, which attempts to model the gravitational interaction as mediated by the affine connection. Its action functional is built up without the use of a metric tensor field by explicitly excluding it from the action.

Despite the fact that an affine model of gravity brings gravitation a step closer to the formulation of gauge theories, it is interesting that the number of possible terms in the action (see Equation (1)) is finite.[7]

Remarkably, even if we start with an affinely connected manifold, without referring to a fundamental metric, there are symmetric $\binom{0}{2}$ tensors derived from the connection that might play the role of metric tensor fields when non-degenerated.

The first connection-descendent object that might be a metric is the symmetric Ricci tensor field [15]. In fact, Einstein and Eddington proposed independently their affine model of gravity by using the square-root of the determinant of the Ricci, i.e., $\sqrt{\text{Ric}}$, as Lagrangian density. Another emergent metric is given by a quadratic combination of the torsion tensor, $g_{\mu\nu}^{(P)} = \mathcal{T}_\mu{}^\lambda{}_\rho \mathcal{T}_\nu{}^\rho{}_\lambda$, which was considered by Popławski in Ref. [27]. We can define a Popławski-like metric by using the $\mathcal{B}$-field, as $g_{\mu\nu}^{(\mathcal{B})} = \mathcal{B}_\mu{}^\lambda{}_\rho \mathcal{B}_\nu{}^\rho{}_\lambda$.

In addition, in three-dimensions, the dual of the $\mathcal{B}$-field, $\mathfrak{B}^{\lambda\rho} = \frac{1}{2}\epsilon^{\mu\nu\lambda}\mathcal{B}_\mu{}^\rho{}_\nu$, is a symmetric $\binom{2}{0}$-tensor density. When this tensor density is non-degenerated, it might be interpreted as an analogous case of the Riemannian $\sqrt{g}g^{\mu\nu}$. Hence, such quantity induces a notion of distance, i.e., introduces a metric.

The general field equations are a clear generalisation of the Einstein equations; in particular, the field equation for the $\mathcal{B}$-field (see Equation (3)) is the analogous case for the standard equations in general relativity, where the following term is provided:

$$2B_1 \mathcal{A}_\nu \mathcal{A}_\rho - 2B_3 \nabla_{(\nu} \mathcal{A}_{\rho)} + 3B_4 \mathcal{B}_\nu{}^\mu{}_\sigma \mathcal{B}_\rho{}^\sigma{}_\mu$$

and it might be interpreted as the energy-momentum tensor field of a geometrically induced matter content. It is worth mentioning that the Ricci tensor field in Equation (3) in general encodes information about nonmetricity, since the symmetric connection, $\Gamma$, is not necessarily compatible with emergent metrics [38,57].

In the sense of the last paragraph, the polynomial affine model of gravity provides nontrivial solutions in the absence of matter, i.e., there are matter-like effects induced by non-Riemannian geometrical quantities.

Lastly, although the Hamiltonian analysis of the polynomial affine model of gravity has not been completed, we have reasons to think that in this model there are propagating degrees of freedom, unlike the three-dimensional version of general relativity.

Before finishing, we would like to remark some of the features that, in four dimensions, differentiate the polynomial affine model of gravity from general relativity.

We know that the four-dimensional version of Polynomial Affine Gravity yields to gravitational interactions (forces) and vacuum solutions that differ from those of general relativity. Nevertheless, our vacuum solutions contain effects that in general relativity can be obtained solely in presence of matter. Ergo, the previously mentioned non-Riemannian matter-like effects might (at least partially) account in the four-dimensional context, for the dark sector of the Universe.

**Funding:** The work OCF and ARZ is sponsored by the "Centro Científico y Tecnológico de Valparaíso" (CCTVal), funded by the Chilean Government through the Centers of Excellence Base Financing Program of Agencia Nacional de Investigación y Desarrollo (ANID), by grant ANID PIA/APOYO AFB180002. This work was funded by ANID Millennium Science Initiative Program ICN2019_044, and benefited from grant PI_LL_19_02 from the Universidad Técnica Federico Santa María. The work of FR is funded by ANID BECAS/DOCTORADO NACIONAL 21211990. BG acknowledges financial support from UTFSM master scholarship No. 034/2021.

**Institutional Review Board Statement:** Not applicable.

**Informed Consent Statement:** Not applicable.

**Data Availability Statement:** This study do not report any data.

**Acknowledgments:** We gratefully acknowledge the constructive comments of anonymous referees, which helped to improve the quality of the final version of the manuscript.

**Conflicts of Interest:** The authors declare no conflict of interest. The funders had no role in the design of the study; in the collection, analyses, or interpretation of data; in the writing of the manuscript; or in the decision to publish the results.

## Notes

[1]   In general relativity, the role of the metric is two-fold: it is not only the geometric object that defines distances but the field is also responsible for the mediation of gravitational interactions.

[2]   In the words of Schrödinger, "For all that I know, no special solution has yet been found which suggests an application to anything that might interest us ..." [19].

[3]   We also refer to these metrics as *emergent* metrics.

[4]   An update on the original ideas in this paper can be found in Ref. [46].

[5]   In previous articles, we refer to this as *dimensional analysis*.

[6]   We shall denote the Popławski metric by Popl or $g^{(P)}$.

[7]   Note that once the metric is included, as in general relativity, the number of possible terms is naturally unbound, since we allow the contraction of the indices that are initially on the same footing. As an example of these terms, we have the scalar curvature, $\mathcal{R}$, and its powers, $\mathcal{R}^n$, opening the window to the $f(\mathcal{R})$-models.

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
