# Peer review of "Polynomial Affine Model of Gravity in Three-Dimensions"

_universe, doi:10.3390/universe8020068_

Round 1

Reviewer 1 Report

The paper is well written and motivated; it is scientifically interesting and presents sound results in the field of modified gravity.

The paper is already good enough in the present format. Even so, I might recommend that the authors emphasize the following. In the paragraph below Eq. (13b), within lines 186-188, they comment on a branch of solutions where Ω = 0 but g and ξ are not determined. The naïve reader will be confused while checking the diagrams containing the decision trees, because these diagram are not consistent with such features. Therefore, I suggest the authors emphasize that the aforementioned branch is not represented in the flow charts (appearing in lines 180 and 181).

Moreover, there is a typo in line 209. Instead of "non-degenarated might be" it should read "non-degenerate it might be". 

I feel the need to point out that the manuscript looks like a reduced version of a paper by the same authors, viz. their Ref. [55]: Castillo-Felisola, O.; Orellana, O.; Perdiguero, J.; Ramírez, F.; Skirzewski, A.; Zerwekh, A.R. Aspects of the Polynomial Affine 318 Model of Gravity in Three dimensions 2021. [arXiv:gr-qc/2107.07209]. I do not see this as plagiarism or other ethical issue under the perspective that the present manuscript was submitted as a Conference Report.

Author Response

Dear referee 1:

Firstly, we would like to thank the exhaust revision of our
manuscript. We also thank the comments to improve the quality of the 
paper.

In response of your suggestions:
- We have modified the decision tree to include the missing branch,
  without the necessity of modifying the text in the former lines 180
  and 181.
- The typo in (former) line 209 has been corrected.

Please note that there additional changes inspired by the comments of
the other referees, included in a highlighted version of the PDF.

Thank you for the suggestions.

Complete list of changes:
- We have added two paragraphs in the introduction (current l.
  46--63) reviewing the known results of the cosmological models in
  the four-dimensional polynomial affine model of gravity.

  It includes two additional footnotes and an extra reference (book by
  Kirill Krasnov). Note that albeit the remaining reference were
  already present in the manuscript, but the order of appearance had
  changed.
- The decision tree for \(\Omega = 0\) has been modified to include
  the case were the condition is satisfied by the fixing the coupling
  constant, and not the defining functions of the connection.
- Change of a grammar mistake (current l. 227): from "...
  non-degenerated might be ..." to "... non-degenerated it might be
  ...".
- At the end of the discussion section we have conclude mentioning
  some of the expected effects in four dimensions, highlighting the
  difference between Polynomial Affine Gravity and General Relativity.
- Addition of funding of one of the authors.
- Inclusion of acknowledgements to the referees.
- Update of the (current) Ref. 57, which has been published.

Reviewer 2 Report

In the manuscript the authors an alternative model of gravity, in which the gravitational properties are described by using a fundamental affine connection, and not by the metric, as in standard general relativity. A general action is introduced in Eq. (1), from which the field equations are derived. Some cosmological applications are also investigated,  by considering the connection homogeneous and isotropic. The weak point of the manuscript is the lack of the presentation of any astrophysical or cosmological applications of the theory. Moreover, no mention of what  new physical effects may be expected, and what would be the implications of the adopted theoretical description for the overall cosmological dynamics. If the authors would discuss some astrophysical/cosmological applications of the presented theory, and what novel features are to be expected from the new gravitational field equations that go beyond standard general relativity, I may recommend this manuscript for publication in Universe.  

Author Response

Dear referee 2:

We would like to thank the exhaust revision of our manuscript. We also
thank the comments to improve the quality of the paper.

We understand the concern regarding the application to astronomical,
astrophysical and cosmological systems, derived from our model. Since
the present manuscript targets a three-dimensional model of gravity,
there are no direct applications on those areas. However, the present
work is part of a project that intends to analyse real world
observations, and thus we have some results on the four-dimensional
version of the polynomial affine gravity.

In order to complete the panorama, and make some contact with the
astrophysical/cosmological observations, we have included some
comments highlighting the difference between General Relativity and
the Polynomial Affine Model of Gravity.

Firstly, in the introduction we added to paragraphs remarking some of
those differences:

Albeit the affine formulations of gravity conceptually differ from
General Relativity, the predictions obtained from earlier models do
not provide significant difference from those of General Relativity,
while their manipulation was harder. It has been understood that even if
two models are equivalent at the dynamical level, their
generalisations might be inequivalent. However, subsequent models
predict novel effects.

In the four-dimensional version of the polynomial affine model of
gravity, even if we restrict ourselves to the torsion-free sector of
the model, the (vacuum) solutions of Polynomial Affine Gravity include
the (vacuum) solutions of General Relativity as a subset. Moreover,
some vacuum solutions of our affine model account for effects that in
General Relativity are induced by the presence of matter. This is
interpreted as a mimicking of matter effects by the non-Riemannian
structure of the geometry. Although the underlying geometry of the
polynomial affine model of gravity does not require the existence of a
metric ab initio, it is possible to define connection-descendent
metrics, allowing to distinguish between null and non-null geodesics
(self-parallel) which would describe the free-fall trajectories for
massless and massive particles. Worth noticing, the existence of a
connection-descendent metric permits to make contact with other
quantities of interest in cosmological and astronomical/astrophysical
applications, such as the red-shift.

Additionally, we modified last paragraph of the discussion section to
remark the four-dimensional effects obtained in Polynomial Affine
Gravity. Explicitly, we complement with the following text:

In the sense of the last paragraph, the polynomial affine model of
gravity provides nontrivial solutions in absence of matter, i.e. there
are matter-like effects induced by non-Riemannian geometrical
quantities.

Lastly, although the Hamiltonian analysis of the polynomial affine
model of gravity has not been finished, we have reasons to think that
in this model there are propagating degrees of freedom, unlike the
three-dimensional version of General Relativity.

Before finishing we would like to remark some of the features that, in
four dimensions, differentiate the polynomial affine model of gravity
from General Relativity. 

We know that the four-dimensional version of Polynomial Affine Gravity
yields to gravitational interactions (forces) and vacuum solutions
that differ from those of General Relativity. Nevertheless, our vacuum
solutions contain effects that in General Relativity can be obtained
solely in presence of matter. Ergo, the previously mentioned
non-Riemannian matter-like effects might (at least partially) account,
in the four-dimensional context, to the dark sector of the Universe.

Please note that there additional changes inspired by the comments of
the other referees, included in a highlighted version of the PDF.

Thank you for the suggestions.

Complete list of changes:
- We have added two paragraphs in the introduction (current l.
  46--63) reviewing the known results of the cosmological models in
  the four-dimensional polynomial affine model of gravity.

  It includes two additional footnotes and an extra reference (book by
  Kirill Krasnov). Note that albeit the remaining reference were
  already present in the manuscript, but the order of appearance had
  changed.
- The decision tree for \(\Omega = 0\) has been modified to include
  the case were the condition is satisfied by the fixing the coupling
  constant, and not the defining functions of the connection.
- Change of a grammar mistake (current l. 227): from "...
  non-degenerated might be ..." to "... non-degenerated it might be
  ...".
- At the end of the discussion section we have conclude mentioning
  some of the expected effects in four dimensions, highlighting the
  difference between Polynomial Affine Gravity and General Relativity.
- Addition of funding of one of the authors.
- Inclusion of acknowledgements to the referees.
- Update of the (current) Ref. 57, which has been published.

Author Response

Dear referee 3:

We would like to thank the exhaust revision of our manuscript.

Please note that there additional changes inspired by the comments of
the other referees, included in a highlighted version of the PDF.

Thank you for the suggestions.

Complete list of changes:
- We have added two paragraphs in the introduction (current l.
  46--63) reviewing the known results of the cosmological models in
  the four-dimensional polynomial affine model of gravity.

  It includes two additional footnotes and an extra reference (book by
  Kirill Krasnov). Note that albeit the remaining reference were
  already present in the manuscript, but the order of appearance had
  changed.
- The decision tree for \(\Omega = 0\) has been modified to include
  the case were the condition is satisfied by the fixing the coupling
  constant, and not the defining functions of the connection.
- Change of a grammar mistake (current l. 227): from "...
  non-degenerated might be ..." to "... non-degenerated it might be
  ...".
- At the end of the discussion section we have conclude mentioning
  some of the expected effects in four dimensions, highlighting the
  difference between Polynomial Affine Gravity and General Relativity.
- Addition of funding of one of the authors.
- Inclusion of acknowledgements to the referees.
- Update of the (current) Ref. 57, which has been published.

Round 2

Reviewer 2 Report

The authors have improved the manuscript, and hence I think that the present version is suitable for publication in Universe.